# Radio Emission from Supernova Remnants: Model Comparison with Observations

**Denis A. Leahy** [1,*] **, Felicity Merrick** [1] **and Miroslav Filipović** [2]

[1] Department of Physics & Astronomy, University of Calgary, Calgary, AB T2N 1N4, Canada
[2] Physical Sciences, Western Sydney University, Locked Bag 1797, Penrith South DC, Sydney, NSW 1797, Australia
* Correspondence: leahy@ucalgary.ca

**Abstract:** Supernova remnants (SNRs) are an integral part in studying the properties of the Galaxy and its interstellar medium. For the current work, we compare the observed radio luminosities of SNRs to predictions based on a recent analytic model applied to 54 SNRs with X-ray observations. We use the X-ray data to determine the properties of shock velocities, ages and circumstellar densities for the SNRs, whereas shock radii are determined from catalogs. With this set of SNR properties, we can calculate the model radio emission and compare it to the observed radio emission for a sample of SNRs. This is the first time that this test has been carried out—previously the SNR properties were assumed instead of derived from X-ray data. With the assumption that the radio emission process depends on SNR properties in the form of power-law functions, we explore ways to improve the radio emission model. The main results of this study are (i) the model has significant deficiencies and cannot reproduce observed radio emission; and (ii) the model can be improved significantly by changing its dependence on SNR parameters, although the improved model is still not accurate. Significant work remains to improve the components of radio emission models, including changes to the SNR evolution model, the radio emitting volume, and the efficiencies for conversion of shock energy into relativistic electrons and for magnetic field amplification.

**Keywords:** supernova remnants; radio emission





## 1. Introduction

Supernova remnants (SNRs) are the outcomes of supernova (SN) explosions and provide key insights regarding the final stage of stellar evolution, the nature of the interstellar medium (ISM) and energy injection into the Galaxy. The basic classifications of SN are Type Ia and Type II, also referred to as core collapse (CC). Type Ia explosions are produced by the thermonuclear explosion of white dwarfs whereas CC explosions are produced by the collapse of the cores of massive stars at the ends of their lives. Both types are highly energetic and release of order $10^{51}$ ergs into the surrounding ISM in the form of kinetic energy of one to several solar masses of matter travelling at thousands of km/s. This sudden injection of energy has a significant effect on the ISM, and understanding how this interaction (the SNR) unfolds can provide insight into the evolution of the ISM and the Galaxy as a whole.

SNRs emit over a wide range of wavelengths by radio through gamma-ray (e.g., [1]). Here the topic of study is a comparison with data of basic analytic models for the radio emission from SNRs. There are observations of nearly three hundred SNRs in the Galaxy [2]. Understanding of their physical conditions requires models to interpret the observations. For example, the age of the SNR is a basic characteristic, and is directly known for only a handful of SNRs which have historical SN records but is derived for other SNRs using models.

SNRs have also been studied by analysis of their associated pulsars, which are young neutron stars rotating, and emitting radio pulsations at their rotation periods. Magnetars,

young neutron stars with extreme surface magnetic fields ($\gtrsim 4 \times 10^{13}$ Gauss) are often clearly associated with SNRs. Only recently have braking indices for magnetars been determined [3], which can be improved with the extra constraint of SNR ages as discussed by [3]. The surface and interior magnetic fields of pulsars are now known to decay over timescales of a few million years, and this decay affects the cooling rate of the neutron star interior. Ref. [4] analyze the high braking index pulsar J1640-4631 using magnetodipole radiation and magnetic field decay models. Ref. [5] applied magnetic decay and temperature evolution calculations to the same pulsar to determine decay model parameters and the pulsar initial rotation period. The high braking indices of two magetars, SGR0501+4516 and 1E2259+586, are analyzed by [6] to find that the double magnetic-dipole moment model, as proposed by [7], can explain the braking indices.

There have been many previous evolutionary models produced for SNRs. They include a simple point explosion in a uniform medium [8], models incorporating radiative energy losses (e.g., [9]), models including effects of the reverse shock in the ejecta at early times [10] and models for the full adiabatic evolution [11]. Ref. [11] incorporates results of hydrodynamic simulations. A number of studies (e.g., [12] for Tycho's SNR) use hydrodynamic simulations customized for a specific SNR. A spherically symmetric model developed by [13,14] incorporates a number of the above models to calculate SNR evolution from the early ejecta-dominated phase through transition to ISM-dominated phase then to radiative phases. For the non-radiative phases, that model uses hydrodynamic simulations to calculate model radii, and temperature and emission measures for the shocked gas of an SNR. These can be compared to the observed radius and temperature and emission measure from X-ray observations to deduce explosion energy, age and interstellar medium density of a given SNR.

The radio emission from SNRs is synchrotron emission from electrons accelerated by the SNR shock. The most recent (and complete) analytic model for radio emission is given by [15], hereafter referred to as the S2017 model. This model uses characteristics such as the shock radius, shock velocity, electron acceleration efficiency, and the magnetic field amplification efficiency to generate a model of the radio luminosity at 1.4 GHz. More complete models have been calculated using hydrodynamic simulations, such as those presented by [16]. Those simulations cover a range of parameters, including ambient density, explosion energy, electron acceleration efficiency and magnetic field amplification efficiency. The radio surface brightness ($\Sigma$, luminosity per unit area) dependence on SNR diameter $D$ is shown to depend on each of the four parameters (Figures 3 and 4 of [16]). The effects of parameter changes are non-linear and different for each parameter. Taken together, the range of parameters can explain the large observed scatter of SNRs in the observed $\Sigma - D$ diagram. The hydrodynamic models are more complex and because of that they are difficult to compare to observed radio emission. The detailed comparison of hydrodynamic simulations for radio emission with observations is an area for future study.

Here we compare the predictions of the S2017 analytic radio emission model with observed SNR radio emission. The SNR variables needed as inputs for the S2017 model are determined using the SNR evolution model of [13]. This paper is organized as follows. We present an updated Table of SNR properties in Section 2, and present the S2017 model in a form suitable for comparison with observed radio emission in Section 3.1. Section 3.2 compares the combined electron acceleration and magnetic field amplification efficiency with values required to match the observed radio emission, to determine what modifications are required. The revised efficiency is presented in Section 3.3, and the results are discussed in Section 4.

## 2. Physical Properties of SNRs

There are 58 Galactic SNRs which have observed X-ray spectra and, from those, forward shock emission measures and shock temperatures. These SNRs were analyzed in two groups to obtain ages, explosion energies and ISM densities: 15 by [17] and 43 by [18]. The results for the 15 SNRs from [17] used an older version of the model [14].

Here SNR properties with the addition of forward shock velocities were calculated using the updated model of [13]. SNR properties were calculated for the *s* = 0 (uniform ISM density), *n* = 7 (power-law index of ejecta density profile) model:

$$\begin{aligned} \rho_{ISM} &= \rho_s r^{-s} \\ \rho_{ej} &\propto r^{-n} \end{aligned} \quad (1)$$

For CC, CC? (uncertain CC-type) and "unk." (unknown) type SNRs the ejecta mass was taken as $5M_\odot$. For Ia and Ia? (uncertain Ia-type) the ejecta mass was taken as $1.4M_\odot$. The errors for shock radius were obtained from the distance uncertainties because angular size uncertainties were negligible. The errors of the observational inputs were propagated through the model to find the errors of the age, energy, density and shock velocity. The SNR properties and errors are given in Table 1 in the Appendix and displayed in Figure 1.

The flux densities at 1.4 GHz for the 58 SNRs were gathered from the catalogue of 295 Galactic SNRs by [2]. Those values were given at 1.0 GHz, so the spectral index of each SNR was used to estimate its 1.4 GHz flux density. If the spectral index of a SNR was unknown, the mean value of all other SNR spectral indices was used to obtain the 1.4 GHz flux density from the 1 GHz value. The resulting 1.4 GHz flux densities are included in Table 1. Four of the 58 SNRs had missing values of flux density, leaving 54 SNRs in our sample for the radio emission analysis.

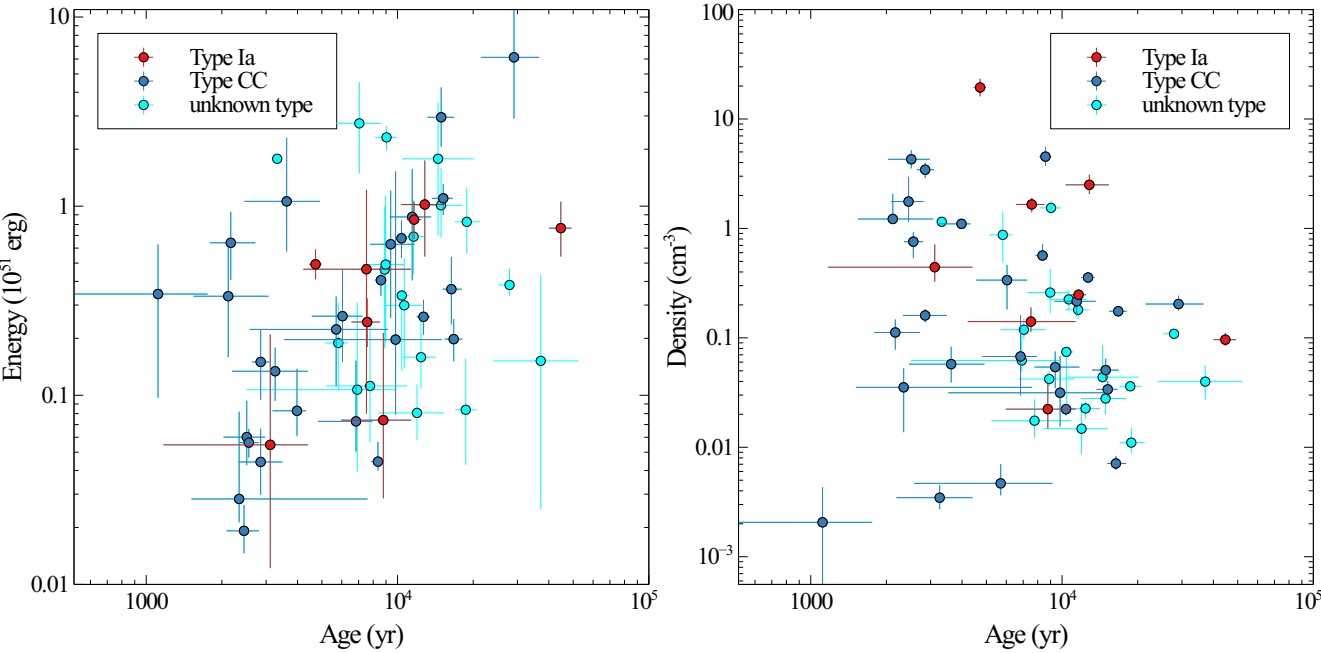

**Figure 1.** Explosion energy (left panel) and density of the ISM (right panel) vs. age for the 58 SNRs using models with *s* = 0, *n* = 7. The Type Ia, Type CC and unknown types are plotted in separate colors. The energies range from $\simeq 2 \times 10^{49}$ erg to $\simeq 8 \times 10^{51}$ erg; the densities range from $\simeq 2 \times 10^{-3}$ cm$^{-3}$ to $\simeq 20$ cm$^{-3}$; and the ages range from $\simeq 1000$ yr to $\simeq 50{,}000$ yr.

Table 1. Observed and Calculated Properties of 58 SNRs.

| Name | Type [1] | Age [$10^2$ yrs] | Energy [$10^{50}$ erg] | Density [$10^{50}$ erg] | Shock Radius [pc] | Shock Velocity [km/s] | Distance [kpc] | 1.4 GHz Flux Density [Jy] [2] |
|---|---|---|---|---|---|---|---|---|
| G 38.7 − 1.3 | CC? | $86^{+60}_{-27}$ | $1.27^{+1.45}_{-0.84}$ | $1.32^{+0.43}_{-0.31}$ | $14.9^{+3}_{-3}$ | $960^{+610}_{-290}$ | $4^{+0.8}_{-0.8}$ | ? [3] |
| G 53.6 − 2.2 | Ia | $446^{+46}_{-45}$ | $7.65^{+2.93}_{-2.24}$ | $9.62^{+1.46}_{-1.19}$ | $34.6^{+3.6}_{-3.6}$ | $310^{+750}_{-20}$ | $7.8^{+0.8}_{-0.8}$ | 6.76 |
| G 67.7 + 1.8 | CC? | $23^{+53}_{-8}$ | $0.28^{+0.54}_{-0.07}$ | $3.53^{+1.73}_{-2.16}$ | $4^{+7}_{-1}$ | $950^{+340}_{-380}$ | $2^{+3.7}_{-0.8}$ | 0.81 |
| G 78.2 + 2.1 | CC? | $94^{+23}_{-16}$ | $6.29^{+5.81}_{-3.73}$ | $5.41^{+2.10}_{-1.94}$ | $19^{+4}_{-4}$ | $910^{+130}_{-13}$ | $2.1^{+0.4}_{-0.4}$ | 270. |
| G 82.2 + 5.3 | unk. | $189^{+25}_{-19}$ | $8.26^{+4.27}_{-2.67}$ | $1.10^{+0.41}_{-0.21}$ | $37^{+4}_{-4}$ | $850^{+60}_{-70}$ | $3.2^{+0.4}_{-0.4}$ | 101. |
| G 84.2 − 0.8 | unk. | $124^{+18}_{-18}$ | $1.59^{+0.96}_{-0.51}$ | $2.26^{+0.34}_{-0.44}$ | $18^{+3}_{-3}$ | $76^{+110}_{-80}$ | $6^{+0.2}_{-0.2}$ | 9.30 |
| G 85.4 + 0.7 | CC? | $57^{+26}_{-22}$ | $1.79^{+0.73}_{-0.59}$ | $2.20^{+1.36}_{-0.51}$ | $12^{+4}_{-4}$ | $1180^{+280}_{-180}$ | $3.5^{+1}_{-1}$ | ? [3] |
| G 85.9 − 0.6 | Ia? | $80^{+19}_{-15}$ | $1.4^{+1.42}_{-0.81}$ | $1.36^{+0.29}_{-0.17}$ | $17^{+5}_{-5}$ | $1010^{+180}_{-140}$ | $4.8^{+1.6}_{-1.6}$ | ? [3] |
| G 89. + 4.7 | CC | $164^{+16}_{-12}$ | $3.64^{+1.75}_{-1.27}$ | $0.71^{+0.12}_{-0.09}$ | $31^{+4}_{-4}$ | $910^{+60}_{-60}$ | $1.9^{+0.3}_{-0.2}$ | 194. |
| G 109.1 − 1.0 | CC | $127^{+80}_{-80}$ | $2.6^{+0.60}_{-0.51}$ | $35.59^{+2.45}_{-2.05}$ | $12.6^{+0.9}_{-0.9}$ | $420^{+20}_{-10}$ | $3.1^{+0.2}_{-0.2}$ | 17.2 |
| G 116.9 + 0.2 | CC | $167^{+14}_{-13}$ | $1.98^{+0.55}_{-0.47}$ | $17.50^{+0.99}_{-0.86}$ | $15.3^{+1.5}_{-1.5}$ | $390^{+20}_{-20}$ | $3.1^{+0.3}_{-0.3}$ | 6.60 |
| G 132.7 + 1.3 | unk. | $279^{+13}_{-27}$ | $3.83^{+0.85}_{-0.48}$ | $10.89^{+0.32}_{-0.30}$ | $24^{+1.2}_{-1.2}$ | $350^{+20}_{-10}$ | $2.1^{+0.1}_{-0.1}$ | 36.8 |
| G 156.2 + 5.7 | CC | $291^{+75}_{-76}$ | $61.2^{+48.3}_{-32.2}$ | $20.38^{+3.91}_{-2.63}$ | $38^{+10}_{-10}$ | $510^{+100}_{-70}$ | $2.5^{+0.8}_{-0.8}$ | 4.22 |
| G 160.9 + 2.6 | CC? | $57^{+35}_{-31}$ | $2.23^{+1.1}_{-1.11}$ | $0.47^{+0.23}_{-0.10}$ | $16^{+8}_{-8}$ | $1570^{+640}_{-290}$ | $0.8^{+0.4}_{-0.4}$ | 88.7 |
| G 166.0 + 4.3 | unk. | $145^{+56}_{-41}$ | $17.8^{+17.4}_{-10.8}$ | $4.37^{+4.25}_{-0.11}$ | $30^{+9}_{-9}$ | $850^{+200}_{-160}$ | $4.5^{+1.5}_{-1.5}$ | 6.18 |
| G 260.4 − 3.4 | CC | $33^{+11}_{-11}$ | $1.34^{+0.45}_{-0.41}$ | $0.35^{+0.10}_{-0.07}$ | $10.5^{+3}_{-3}$ | $1800^{+330}_{-690}$ | $1.3^{+0.3}_{-0.3}$ | 110. |
| G 272.2 − 3.2 | Ia | $75^{+38}_{-33}$ | $4.64^{+7.56}_{-3.84}$ | $14.1^{+4.94}_{-2.71}$ | $14^{+7}_{-7}$ | $760^{+340}_{-200}$ | $6^{+4}_{-4}$ | 0.33 |
| G 296.7 − 0.9 | unk. | $116^{+13}_{-13}$ | $6.9^{+3.85}_{-2.52}$ | $18.0^{+4.83}_{-3.28}$ | $17^{+2}_{-2}$ | $610^{+50}_{-50}$ | $10^{+0.9}_{-0.9}$ | 2.54 |
| G 296.8 − 0.3 | CC? | $104^{+6}_{-5}$ | $6.76^{+1.65}_{-1.46}$ | $2.23^{+0.26}_{-0.27}$ | $23.7^{+1.5}_{-1.5}$ | $1050^{+40}_{-40}$ | $9.6^{+0.6}_{-0.6}$ | 7.35 |
| G 299.2 − 2.9 | Ia | $88^{+26}_{-28}$ | $0.74^{+1.4}_{-0.45}$ | $2.23^{+1.60}_{-0.75}$ | $14^{+5}_{-5}$ | $770^{+280}_{-130}$ | $5^{+1}_{-1}$ | 0.43 |
| G 304.6 + 0.1 | unk. | $89^{+23}_{-21}$ | $4.62^{+5.34}_{-2.84}$ | $4.21^{+0.77}_{-0.62}$ | $18^{+6}_{-6}$ | $960^{+220}_{-140}$ | $15^{+5}_{-5}$ | 11.8 |
| G 306.3 − 0.9 | Ia | $128^{+25}_{-25}$ | $10.2^{+7.25}_{-4.79}$ | $250.16^{+59.75}_{-43.83}$ | $11.6^{+2.3}_{-2.3}$ | $310^{+60}_{-40}$ | $20^{+4}_{-4}$ | 0.14 |
| G 308.4 − 1.4 | CC? | $29^{+60}_{-5}$ | $0.44^{+0.22}_{-0.15}$ | $16.03^{+2.30}_{-2.18}$ | $4.1^{+0.5}_{-0.5}$ | $800^{+100}_{-90}$ | $3.1^{+0.3}_{-0.3}$ | 0.34 |
| G 309.2 − 0.6 | CC? | $11^{+60}_{-6}$ | $3.43^{+2.85}_{-2.46}$ | $0.21^{+0.23}_{-0.15}$ | $8^{+4}_{-4}$ | $4010^{+1570}_{-1220}$ | $2.8^{+0.2}_{-0.2}$ | 6.12 |
| G 311.5 − 0.3 | unk. | $69^{+29}_{-44}$ | $1.07^{+2.01}_{-0.68}$ | $6.21^{+4.45}_{-1.33}$ | $10^{+6}_{-6}$ | $810^{+440}_{-200}$ | $10^{+5}_{-5}$ | 2.54 |
| G 315.4 − 2.3 | Ia | $116^{+8}_{-8}$ | $8.49^{+2.05}_{-1.79}$ | $24.75^{+2.79}_{-2.71}$ | $17^{+1.1}_{-1.1}$ | $580^{+30}_{-30}$ | $2.8^{+0.2}_{-0.2}$ | 40.0 |
| G 322.1 + 0.0 | CC | $43^{+28}_{-20}$ | $1.4^{+3.65}_{-0.89}$ | $4.99^{+2.19}_{-1.41}$ | $8.5^{+0.8}_{-0.8}$ | $1110^{+490}_{-270}$ | $9.3^{+0.9}_{-0.9}$ | ? [3] |
| G 327.4 + 0.4 | unk. | $106^{+19}_{-17}$ | $2.99^{+2.34}_{-1.59}$ | $22.47^{+4.83}_{-4.47}$ | $13.1^{+2.7}_{-2.7}$ | $540^{+70}_{-60}$ | $4.3^{+0.8}_{-0.8}$ | 24.5 |
| G 330.0 + 15 | unk. | $371^{+151}_{-131}$ | $1.52^{+2.82}_{-1.27}$ | $3.98^{+1.66}_{-1.27}$ | $27^{+12}_{-12}$ | $310^{+100}_{-80}$ | $1^{+0.5}_{-0.5}$ | 296. |
| G 330.2 + 1.0 | CC | $98^{+52}_{-63}$ | $1.97^{+13.3}_{-1.18}$ | $3.15^{+3.69}_{-1.60}$ | $16^{+8}_{-8}$ | $860^{+550}_{-240}$ | $10^{+5}_{-5}$ | 4.52 |
| G 332.4 − 0.4 | CC? | $40^{+3}_{-8}$ | $0.83^{+0.55}_{-0.22}$ | $110.05^{+10.10}_{-7.06}$ | $4.5^{+0.9}_{-0.9}$ | $630^{+60}_{-50}$ | $3^{+0.3}_{-0.3}$ | 23.7 |
| G 332.4 + 0.1 | unk. | $70^{+16}_{-14}$ | $27.4^{+17.9}_{-12.5}$ | $11.87^{+02.98}_{-2.19}$ | $20^{+4}_{-4}$ | $1180^{+180}_{-140}$ | $9.2^{+1.7}_{-1.7}$ | 22.0 |
| G 332.5 − 5.6 | unk. | $120^{+33}_{-35}$ | $0.81^{+0.33}_{-0.23}$ | $1.48^{+1.07}_{-0.61}$ | $15.5^{+4}_{-4}$ | $720^{+120}_{-70}$ | $3^{+0.8}_{-0.8}$ | 1.5 |
| G 337.2 − 0.7 | Ia? | $31^{+13}_{-19}$ | $0.55^{+1.55}_{-0.43}$ | $44.23^{+27.15}_{-11.65}$ | $4.7^{+3}_{-3}$ | $760^{+540}_{-210}$ | $5.5^{+3.5}_{-3.5}$ | 1.31 |
| G 337.8 − 0.1 | CC? | $36^{+13}_{-12}$ | $10.6^{+12.4}_{-4.85}$ | $5.75^{+2.55}_{-1.84}$ | $13.4^{+1.3}_{-1.3}$ | $1930^{+520}_{-400}$ | $12.3^{+1.2}_{-1.2}$ | 12.7 |
| G 347.3 − 0.5 | CC | $68^{+11}_{-20}$ | $0.73^{+0.80}_{-0.22}$ | $6.75^{+9.40}_{-3.78}$ | $8.8^{+1.7}_{-1.7}$ | $720^{+17}_{-80}$ | $1^{+0.2}_{-0.2}$ | 25.5 |
| G 348.5 + 0.1 | CC | $114^{+22}_{-21}$ | $8.77^{+6.96}_{-4.72}$ | $21.57^{+5.81}_{-5.61}$ | $17.2^{+3.5}_{-3.5}$ | $620^{+90}_{-100}$ | $7.9^{+1.6}_{-1.6}$ | 65.1 |
| G 348.7 + 0.3 | CC | $149^{+19}_{-18}$ | $29.5^{+13}_{-8.87}$ | $5.07^{+1.38}_{-0.93}$ | $32.8^{+3.3}_{-3.3}$ | $890^{+70}_{-60}$ | $13.2^{+1.3}_{-1.3}$ | 23.5 |
| G 349.7 + 0.2 | CC | $29^{+2}_{-2}$ | $1.5^{+0.74}_{-0.56}$ | $343.79^{+54.59}_{-55.68}$ | $3.7^{+0.4}_{-0.4}$ | $650^{+40}_{-40}$ | $11.5^{+1.2}_{-1.2}$ | 16.9 |
| G 350.1 − 0.3 | CC | $25^{+5}_{-5}$ | $0.60^{+0.34}_{-0.17}$ | $428.43^{+90.78}_{-74.91}$ | $2.6^{+0.5}_{-0.5}$ | $580^{+70}_{-50}$ | $9^{+1.8}_{-1.8}$ | 4.58 |
| G 352.7 − 0.1 | Ia? | $76^{+9}_{-10}$ | $2.44^{+0.81}_{-0.64}$ | $165.61^{+24.59}_{-25.27}$ | $7.6^{+0.5}_{-0.5}$ | $400^{+40}_{-50}$ | $7.5^{+0.5}_{-0.5}$ | 3.27 |
| G 355.6 − 0.0 | unk. | $90^{+17}_{-17}$ | $4.91^{+6.45}_{-3}$ | $25.91^{+16.79}_{-9.18}$ | $13.2^{+2.7}_{-2.7}$ | $630^{+100}_{-90}$ | $13^{+2.6}_{-2.6}$ | 2.55 |
| G 359.1 − 0.5 | unk. | $187^{+20}_{-17}$ | $0.84^{+0.73}_{-0.41}$ | $3.61^{+0.34}_{-0.17}$ | $17.5^{+3.5}_{-3.5}$ | $110^{+120}_{-30}$ | $5^{+1}_{-1}$ | 12.2 |
| G 18.1 − 0.1 | unk. | $58^{+5}_{-7}$ | $1.89^{+1.2}_{-0.83}$ | $87.3^{+55.7}_{-38.4}$ | $7^{+0.2}_{-0.2}$ | $550^{+80}_{-60}$ | $6.4^{+0.2}_{-0.2}$ | 3.89 |
| G 21.5 − 0.9 | CC | $25^{+4}_{-4}$ | $0.19^{+0.07}_{-0.05}$ | $176^{+120}_{-61}$ | $2.1^{+0.1}_{-0.1}$ | $480^{+40}_{-40}$ | $4.4^{+0.2}_{-0.2}$ | 5.96 |
| G 21.8 − 0.6 | unk. | $104^{+7}_{-3}$ | $3.37^{+0.23}_{-2.02}$ | $7.44^{+0.43}_{-5.3}$ | $16.3^{+0.6}_{-0.6}$ | $710^{+250}_{-80}$ | $5.6^{+0.2}_{-0.2}$ | 53.8 |
| G 27.4 + 0.0 | CC | $21^{+10}_{-6}$ | $3.34^{+2.96}_{-1.75}$ | $122^{+85}_{-1.25}$ | $4.6^{+0.25}_{-0.25}$ | $1110^{+220}_{-290}$ | $5.8^{+0.3}_{-0.3}$ | 4.78 |
| G 28.6 − 0.1 | unk. | $149^{+32}_{-30}$ | $10.1^{+5.8}_{-3.28}$ | $2.79^{+0.79}_{-0.79}$ | $29.3^{+0.9}_{-0.9}$ | $840^{+110}_{-100}$ | $9.6^{+0.3}_{-0.3}$ | 2.55 |

**Table 1.** *Cont.*

| Name | Type [1] | Age [$10^2$ yrs] | Energy [$10^{50}$ erg] | Density [$10^{50}$ erg] | Shock Radius [pc] | Shock Velocity [km/s] | Distance [kpc | 1.4 GHz Flux Density [Jy] [2] |
|---|---|---|---|---|---|---|---|---|
| G 29.7 − 0.3 | CC | $26^{+2}_{-2}$ | $0.56^{+0.10}_{-0.09}$ | $75.7^{+17}_{-22.2}$ | $3.3^{+0.15}_{-0.15}$ | $720^{+40}_{-40}$ | $5.6^{+0.3}_{-0.3}$ | 8.09 |
| G 31.9 + 0.0 | CC | $86^{+5}_{-5}$ | $4.06^{+0.91}_{-0.7}$ | $453^{+104}_{-80}$ | $7.2^{+0.4}_{-0.4}$ | $1070^{+20}_{-20}$ | $7.1^{+0.4}_{-0.4}$ | 21.3 |
| G 32.8 − 0.1 | unk. | $78^{+31}_{-25}$ | $1.12^{+1.35}_{-0.55}$ | $1.75^{+0.99}_{-0.52}$ | $13^{+0.8}_{-0.8}$ | $930^{+240}_{-170}$ | $4.8^{+0.3}_{-0.3}$ | 10.3 |
| G 33.6 + 0.1 | CC | $84^{+4}_{-5}$ | $0.45^{+0.12}_{-0.05}$ | $56.5^{+15.3}_{-4.6}$ | $6.3^{+0.4}_{-0.4}$ | $390^{+20}_{-30}$ | $3.5^{+0.3}_{-0.3}$ | 16.8 |
| G 34.7 − 0.4 | unk. | $90^{+9}_{-9}$ | $23.1^{+3.4}_{-3.4}$ | $154^{+26}_{-21}$ | $13^{+1.3}_{-1.3}$ | $580^{+40}_{-30}$ | $3^{+0.3}_{-0.3}$ | 212. |
| G 39.2 − 0.3 | CC | $60^{+12}_{-15}$ | $2.62^{+1.97}_{-1.12}$ | $33.7^{+12.7}_{-15.5}$ | $9.1^{+0.5}_{-0.5}$ | $710^{+160}_{-90}$ | $8.5^{+0.5}_{-0.5}$ | 16.1 |
| G 41.1 − 0.3 | Ia | $47^{+3}_{-3}$ | $4.93^{+0.97}_{-0.82}$ | $1940^{+390}_{-330}$ | $4.45^{+0.25}_{-0.25}$ | $300^{+20}_{-20}$ | $8.5^{+0.5}_{-0.5}$ | 21.1 |
| G 43.3 − 0.2 | unk. | $33^{+1}_{-1}$ | $17.8^{+1}_{-1}$ | $115^{+7}_{-5}$ | $8.55^{+0.3}_{-0.3}$ | $1090^{+20}_{-20}$ | $11.3^{+0.4}_{-0.4}$ | 32.6 |
| G 49.2 − 0.7 | CC | $152^{+14}_{-15}$ | $11.^{+2.1}_{-0.5}$ | $3.38^{+0.66}_{-0.5}$ | $29^{+3.3}_{-3.3}$ | $800^{+60}_{-40}$ | $5.6^{+0.6}_{-0.6}$ | 145. |
| G 54.1 + 0.3 | CC | $22^{+6}_{-4}$ | $6.4^{+2.93}_{-2.33}$ | $11.2^{+3.5}_{-3.41}$ | $7.9^{+1.3}_{-1.3}$ | $2040^{+230}_{-250}$ | $4.9^{+0.8}_{-0.8}$ | 0.48 |

[1] CC (core collapse type), CC? (uncertain CC type) and "unk." (unknown type) SNRs are taken to have $5M_\odot$ ejecta mass; Type Ia and Type Ia? (uncertain Type Ia) are taken to have $1.4M_\odot$ ejecta mass. [2] 1 Jy is $10^{-23}$ erg cm$^{-2}$ s$^{-1}$ Hz$^{-1}$. [3] "?" denotes SNRs without measured 1.4 GHz flux density: these are omitted for the radio analysis.

## 3. Analysis

### 3.1. Model for Radio Emission at 1.4 GHz

The radio emission model of [15] (referred to as the S2017 model) is analyzed here. The S2017 model gives the radio luminosity density at 1.4 GHz ($L_{1.4}$):

$$L_{1.4} \approx \left(2.2 \times 10^{24} \text{ ergs/s/Hz}\right) \left(\frac{R_s}{10 \text{ pc}}\right)^3 \left(\frac{\epsilon_e}{10^{-2}}\right) \left(\frac{\epsilon_b^u}{10^{-2}}\right)^{0.8} \left(\frac{v_s}{500 \text{ km/s}}\right)^{3.2} \quad (2)$$

where $R_s$ is the shock radius, $\epsilon_e$ is the fraction of shock energy from relativistic electrons, referred to as the electron acceleration efficiency, $\epsilon_b^u$ is the fraction of shock energy in an amplified upstream magnetic field, referred to as the magnetic field amplification efficiency, and $v_s$ is the shock velocity. The magnetic field amplification efficiency is calculated following [15]:

$$\epsilon_b^u = \frac{\xi_{cr}}{2}\left(\frac{v_s}{c} + \frac{1}{M_A}\right) \quad (3)$$

where $\xi_{cr}$ is the cosmic ray acceleration efficiency and $M_A$ is the Alfven Mach number. The value of $\xi_{cr} = 0.1$ was used in S2017. The lowest $M_A$ for our set of SNRs was 29, so that the low $M_A$ formula for $\xi_{cr}$ was not needed. The standard formula for $M_A = v_s \frac{\sqrt{4\pi\rho_0}}{B_0}$ is used, with $B_0$ dependence on density taken from [19] with exponent of 0.5 (consistent with their range $0.47 \pm 0.08$). To relate number $n_H$ and mass density $\rho_0$ of the ISM, we use $\rho_0 = \mu m_H n_H$ with $\mu = 1.356$ for solar abundances [20].

We rearrange Equation (2) to yield the radio flux density at 1.4 GHz:

$$F_{1.4} \approx \left(\frac{2.2 \times 10^{24}}{4\pi} \text{ ergs/s/Hz}\right) \theta^3 \left(\frac{d}{(10 \text{ pc})^3}\right) \left(\frac{\epsilon_e}{10^{-2}}\right) \left(\frac{\epsilon_b^u}{10^{-2}}\right)^{0.8} \left(\frac{v_s}{500 \text{ km/s}}\right)^{3.2} \quad (4)$$

where $\theta$ is the angular size of the supernova remnant and $d$ is the distance to the supernova remnant. By using angular size, which is well measured, rather than shock radius we reduce the impact of errors of distance.

The predicted flux densities of the SNRs for $\epsilon_e = 4.2 \times 10^{-3}$ used in the S2017 model, are compared to the observed flux densities in Figure 2. The predicted values are poorly correlated with the observed ones, with predicted flux densities scattered over a factor of $10^6$, compared to a $\sim 10^3$ range for the observed values. This implies that the model is missing important factors or that the model can be improved by changing some of the assumptions.

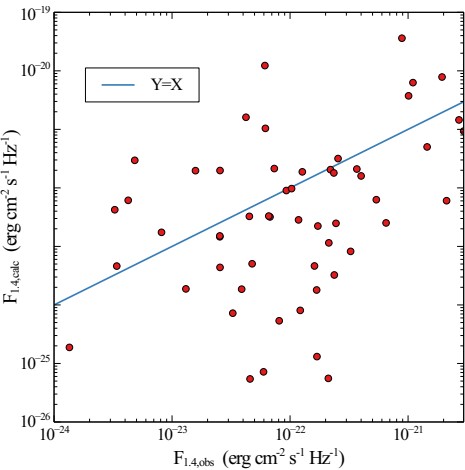

**Figure 2.** Calculated flux densities using the S2017 model and SNR calculated properties compared to observed flux densities for the 54 SNRs with measured flux densities. The line is that for equal calculated and observed flux densities. This plot illustrates the large scatter and lack of agreement between the model and observed flux densities.

To check that the S2017 model can produce the same range of radio luminosities as the observed one, we calculate the S2017 luminosity vs. age by using the shock velocity and shock radius vs. age from the SNR model of [13]. The explosion energy was fixed at $10^{51}$ erg, the local interstellar density was fixed at 1 cm$^{-3}$ and the ejecta mass was 1.4 or $5 M_\odot$. The combined efficiency, $\eta$, is defined as:

$$\eta = \left( \frac{\epsilon_e}{10^{-2}} \right) \left( \frac{\epsilon_b^u}{10^{-2}} \right)^{0.8} \tag{5}$$

Three values for $\eta$ were used to produce the set of radio luminosity evolution curves in Figure 3.

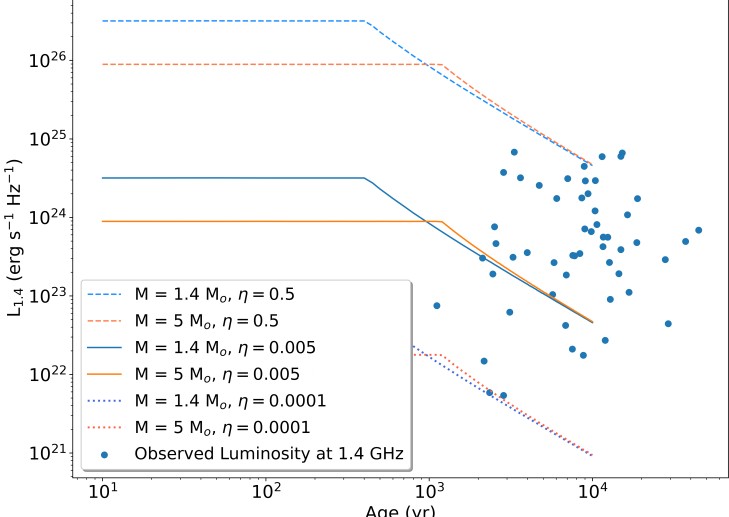

**Figure 3.** Predicted radio luminosity evolution from the S2017 model for several different sets of input parameters (lines), compared to observed luminosity for the 54 SNRs in the sample. The ages (for both models and observations) and the shock velocities and radii (for the models) were calculated with $E$ of $10^{51}$ erg and $n_0$ of 1 cm$^{-3}$. M = 1.4 curves correspond to Ia SNRs. M = 5 curves correspond to CC SNRs. $\eta$ is the efficiency defined by Equation (5).

From Figure 3, it is seen that $\eta$ has a range of approximately three orders of magnitude. If $\epsilon_e$ is approximated to be equal to $\epsilon_b^u$, this gives us $\epsilon_e \sim 10^{-4}$ to $10^{-2}$. These values can be compared to $\epsilon_e = 4.2 \times 10^{-3}$ chosen in the S2017 model.

### 3.2. Analysis of the Efficiency

To examine how to improve the S2017 model, we start by analysis of the efficiency. We test whether a dependence of the efficiency on SNR properties can improve the agreement between model and observations. The combined efficiency $\eta$ can be inserted into Equation (4) to yield:

$$F_{1.4} \approx \left( \frac{2.2 \times 10^{24}}{4\pi} \text{ ergs/s/Hz} \right) \theta^3 \left( \frac{d}{(10 \text{ pc})^3} \right) \eta \left( \frac{v_s}{500 \text{ km/s}} \right)^{3.2} \quad (6)$$

The combined efficiency $\eta$ required to reproduce the observed radio emission using Equation (6) is calculated and then compared to SNR properties. The $\eta$ values from the observed radio flux densities are shown vs. shock velocity, density, and shock radius. This revealed the trends shown in Figure 4. The fit lines are a least-squares linear regression (in log-log space), corresponding to the best fit power-law relation for the variables:

$$Y = C1(X/C2)^u \quad (7)$$

where $C1$, $C2$ and $u$ are constants, with $u$ the power-law index, $C1$ the normalization for the fit to $Y$, and $C2$ chosen as the scale for $X$. For $v_s$, the scale was chosen as 500 km/s, for $n_0$ the scale was chosen as 1 cm$^{-3}$, and for $R_s$ the scale was chosen as 10 pc. Figure 4 shows that $\eta$ is approximately proportional to $v_s^{-3}$, $n_0^1$, and $R_s^{-21}$. This indicates a dependence on the density of the ISM which is not present in the S2017 model, as well as different dependences on $v_s$ and $R_s$.

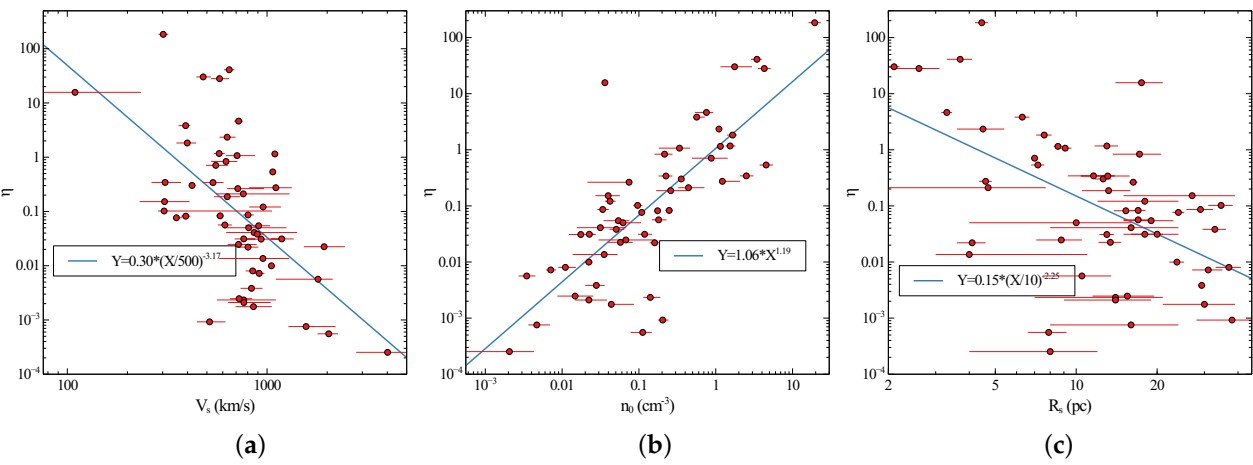

**Figure 4.** (**a**) $\eta$ vs. shock velocity $v_s$ of each SNR. (**b**) $\eta$ vs. density $n_0$. (**c**) $\eta$ vs. shock radius $R_s$. The best-fit power-law line is given in each case.

The shock radius is global property of each SNR, and should have no effect on the efficiency which should depend on local variables. The relation between $\eta$ and the shock radius could be an indirect consequence of the $\eta$ dependence on the shock velocity and ISM density. Alternately, the assumption of the S2017 model that the volume of the radio emitting region grows as $R_s^3$ might not be accurate.

### 3.3. Efficiency Dependence on Shock Velocity and ISM Density

A further analysis is carried out to better determine how the efficiency, $\eta$ depends on $v_s$, $n_0$ and $R_s$. The best fit power-law indices for the dependence of $\eta$ on $v_s$, $n_0$ and $R_s$ are $-3.17$, $1.19$ and $-2.25$ with corresponding correlation coefficients ($r^2$ values) for the three

fits of 0.35, 0.61 and 0.29, respectively. If an improved dependence of $\eta$ on the physical parameters of each SNR is accounted for, there should be no residual dependence on the parameters, or at least a significantly reduced dependence. This implies that a power-law fit to the residual dependence should be nearly flat (power-law index near 0) and the correlation coefficients should be reduced.

The correlation between $\eta$ and shock velocity and density, seen in Figure 4, is modelled by introducing an explicit power-law dependence:

$$\eta = H(n_0/\mathrm{cm}^{-3})^{\alpha} (v_s/500 \text{ km/s})^{\beta} \tag{8}$$

where $H$ is new parameter. The values of $\alpha$ and $\beta$ were varied to achieve smallest dependences of $H$ on shock velocity and density[2]. The best fit values are $\alpha \simeq 0.85$ and $\beta \simeq -1.9$, i.e., $\eta$ is approximately proportional to $n_0^{0.85}$ and $v_s^{-1.9}$.

The above values of $\alpha$ and $\beta$ result in the relations between $H$ and shock velocity and density shown in Figure 5. There is no significant remaining correlation of $H$ with density and velocity, as expected. The best fit power-law slopes of $H$ vs. $v_s$ and $n_0$ are 0.008 and 0.11, respectively. By introducing the density and velocity dependences, the dependence of $H$ with shock radius is reduced from a power-law with index $-2.25$ to a power-law with index $-0.96$. The resulting correlation coefficients for $H$ vs. $v_s$, $n_0$ and $R_s$ are significantly smaller: $7 \times 10^{-6}$, 0.016 and 0.16, respectively. The scatter of $H$ vs. $n_0$, $v_s$ and $R_s$ is reduced: from full range of $\sim 10^5$ for $\eta$ (Figure 4) to $\sim 5 \times 10^3$ for $H$ (Figure 5).

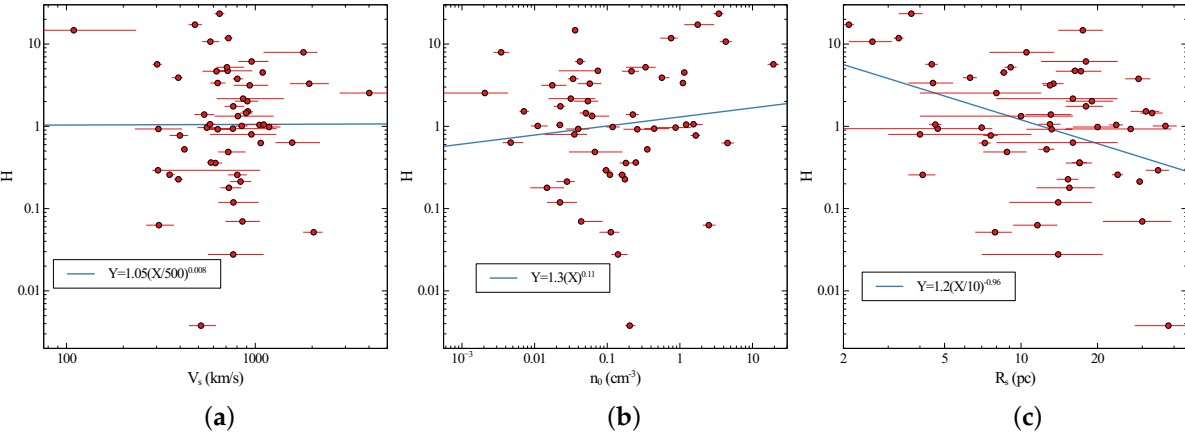

**Figure 5.** (**a**) $H$ vs. shock velocity $v_s$ of each SNR. (**b**) $H$ vs. density $n_0$. (**c**) $H$ vs. shock radius $R_s$. The best-fit power-law line is given in each case.

The net resulting formula for the combined efficiency obtained from the above analysis is Equation (8) with $\alpha = 0.85$ $\beta = -1.9$. The best fit constant $H$ is 1.1 for $n_0$ in units of $\mathrm{cm}^{-3}$ and $v_s$ in units of 500 km/s.

$$\eta = 1.1 \, (n_0/\mathrm{cm}^{-3})^{0.85} (v_s/500 \text{ km/s})^{-1.9} \tag{9}$$

This revised combined efficiency (Equations (5) and (9)) when substituted into the luminosity formula yields a new empirical luminosity formula:

$$L_{1.4} \approx \left(2.4 \times 10^{24} \text{ ergs/s/Hz}\right) \left(\frac{R_s}{10 \text{ pc}}\right)^3 (n_0/\mathrm{cm}^{-3})^{0.85} \left(\frac{v_s}{500 \text{ km/s}}\right)^{1.3} \tag{10}$$

### 3.4. Exploring Dependence on Shock Radius

The efficiency $\eta$ should depend only on local shock properties and not on $R_s$. However, the dependencies of $\eta$ and $H$ on $R_s$ lead us to conclude that the assumption that the radio emitting volume depends on $R_s^3$ is probably incorrect. Thus, we take one more step and rewrite $\eta$ as:

$$\eta = J(n_0/\mathrm{cm}^{-3})^\alpha \, (v_s/500 \, \mathrm{km/s})^\beta \, (R_s/10 \, \mathrm{pc})^\gamma \tag{11}$$

where $J$ is new parameter. Similar to above, in Figure 6, we find the values of $\alpha$, $\beta$ and $\gamma$ which give minimum variation of $J$ with $n_0$, $v_s$ and $R_s$. The values obtained are $\alpha = 0.85$, $\beta = 1.9$ and $\gamma = -0.9$. The best-fit power-law fits for $J$ vs. $n_0$, $v_s$ and $R_s{}^3$ have power-law slopes of $-0.08$, $-0.04$ and $0.09$, with correlation coefficients of $0.01$, $2 \times 10^{-4}$ and $7 \times 10^{-4}$.

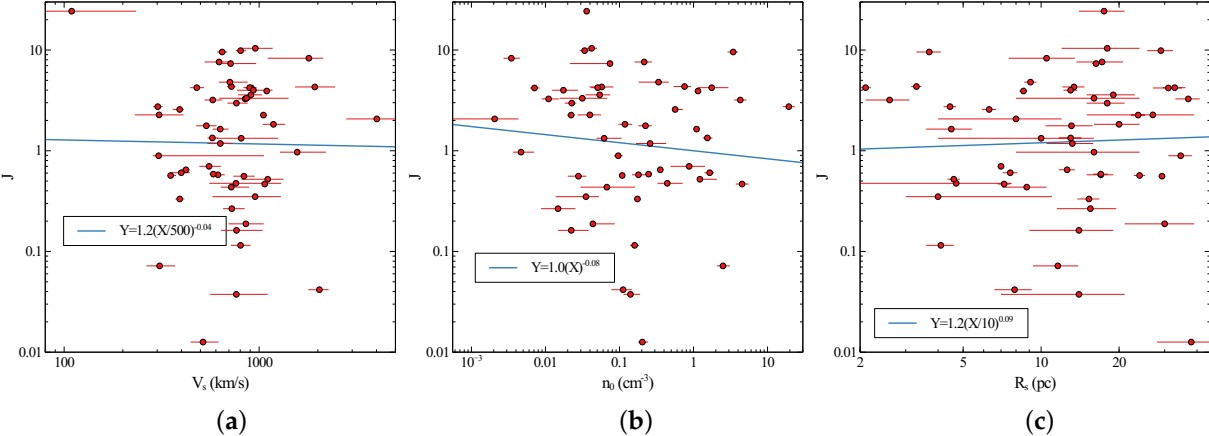

**Figure 6.** (**a**) $J$ vs. shock velocity $v_s$ of each SNR. (**b**) $J$ vs. density $n_0$. (**c**) $J$ vs. shock radius $R_s$. The best-fit power-law line is given in each case.

The best fit constant $J$ is 1.1 for $n_0$ in units of $\mathrm{cm}^{-3}$, $v_s$ in units of 500 km/s, and $R_s$ in units of 10 pc. The net resulting formula for the combined efficiency obtained from the above analysis is:

$$\eta = 1.1 \, (n_0/\mathrm{cm}^{-3})^{0.85} \, (v_s/500 \, \mathrm{km/s})^{-1.9} \, (R_s/10 \, \mathrm{pc})^{-0.9} \tag{12}$$

Because this is obtained from comparing the model to the observations, it is no longer the efficiency of shock acceleration and magnetic field amplification: it includes factors to compensate for missing dependencies in the model luminosity of Equation (2). This revised combined efficiency (Equation (11)) when substituted into the luminosity formula yields a new empirical luminosity formula:

$$L_{1.4} \approx \left(2.3 \times 10^{24} \, \mathrm{ergs/s/Hz}\right) \left(\frac{R_s}{10 \, \mathrm{pc}}\right)^{2.1} (n_0/\mathrm{cm}^{-3})^{0.85} \left(\frac{v_s}{500 \, \mathrm{km/s}}\right)^{1.3} \tag{13}$$

In summary, we find an empirical relationship between the observed 1.4 GHz luminosity of a SNR and the three SNR parameters (density of the local ISM, shock velocity and shock radius). This is based on the assumption that the dependencies of radio luminosity on the SNR parameters of shock velocity, ambient density and radius are power-law functions.

## 4. Discussion

The model radio emission for a large sample of SNRs calculated using the physical properties of individual SNRs has not been presented previously. TheSNR properties have been enabled recently by the extensive evolutionary models of [13]. The model for radio emission of [15] (the S2017 model) was used because it is the most recent and complete analytic model for radio emission from SNRs. It relates the radio luminosity at 1.4 GHz to the physical properties of a SNR: shock radius $R_s$, electron acceleration efficiency $\epsilon_e$, magnetic field amplification efficiency $\epsilon_b$, and shock velocity $v_s$. The assumptions include the emitting volume is proportional to $R_s^3$ and $\epsilon_e$ is constant. A reasonable model for $\epsilon_b$ is given (Appendix A of [15]), which incorporates the ISM magnetic field from [19][4], and analytic approximations for non-resonant [21] and resonant [22] modes for field amplification.

A detailed test of this model against observations is carried out for the first time in this study. We use the sample of 54 Galactic SNRs for which there are X-ray observations, which allow the calculation of SNR properties required to predict the radio emission, and for which there are radio observations. This investigation has revealed that the S2017 model, which is summarized above in Equation (2), does not reproduce the observed radio emission of SNRs. The range of likely input parameters results in a range of radio luminosities comparable to those observed, as shown in Figure 3; however, the comparison of flux densities for individual SNRs in Figure 2 shows that the model fails for individual SNRs.

The current study has revealed a dependence on the density of the local interstellar medium which is not present in the S2017 model. Inclusion of a density dependence and different shock velocity and radius dependences, as given by Equations (8) or (11) here, significantly improves the agreement between the altered model and the observations.

The observed SNR luminosities are compared to the model luminosities in Figure 7. The models shown include the S2017 model, our empirical model with dependence of $\eta$ on $n_0$ and $v_s$ (Equation (9), called model a), and with dependence of $\eta$ on $n_0$, $v_s$ and $R_s$ (Equation (12), called model b). The deviations of predicted luminosities from observed luminosities is still large for the new empirical models proposed here. We have quantified this by using power-law fits of model to observed luminosities for the three models. For the S2017 model, the best fit power-law has index of 0.26 and correlation coefficient of 0.03. Model a has a best fit power-law has index of 0.45 and a correlation coefficient of 0.25. Model b has a best fit power-law has index of 0.36 and a correlation coefficient of 0.27. The empirical model a and model b compare more favorably with observed luminosities, but they are still inadequate. A main point here is that new analytic models beyond that of S2017 need to be developed before the radio emission from SNRs is understood well enough to link radio emission to SNR properties.

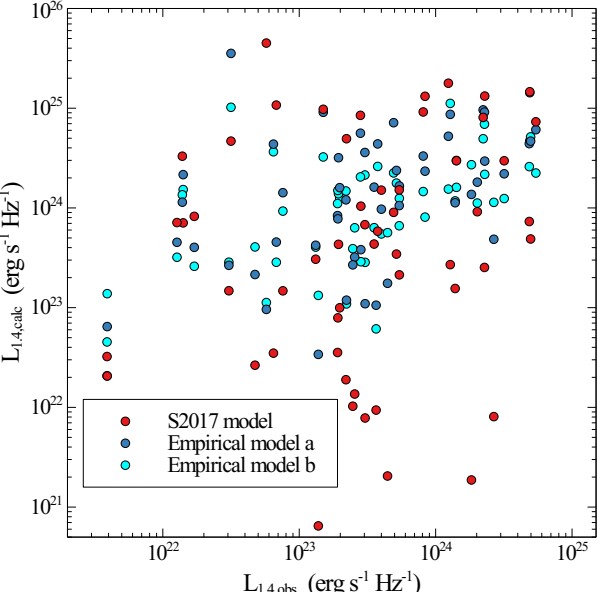

**Figure 7.** Predicted 1.4 GHz luminosities vs. observed luminosities for the sample of 54 SNRs. The S2017 model luminosities are shown in red, the empirical model a presented here in blue and the empirical model B in green. The S2017 model has the most outliers from equality of observed and model fluxes, model a has an intermediate number of outliers and model b has fewest outliers.

One main assumption of the empirical models here (represented by Equations (6), (8) and (11)) is that the dependences are power-law in form. The dependence of ISM magnetic field on density used by S2017 was derived from [19], who found the relation in a study of molecular clouds with densities of $10^2$ to $10^7$ cm$^{-3}$, whereas SNRs have ISM densities of $10^{-3}$ to 30 cm$^{-3}$. Thus, the relation from [19] might not be applicable. If the relation

of magnetic field, and $\epsilon_b$, on $n_0$ is power-law, it is represented in our empirical model, but if it is not power-law another form needs to be considered. In fact, simple analytical models may not be adequate to model the radio emission from SNRs, as explored in the hydrodynamical studies of [16].

Another factor that could cause some of the differences between model radio emission and observed radio emission is stochastic variations in ISM magnetic field. The differences in initial ISM magnetic field for different SNRs would result in differences in amplified magnetic field and thus differences in the parameter $\epsilon_b^u$ for each SNR (see Equation (2), and discussion in [15]).

Previous studies of radio emission have hinted that the ISM density may be important for radio emission. The statistical study of radio emission from SNRs by [23] found that earlier models using constant efficiencies of particle acceleration and magnetic field amplification do not fit the data well. In addition, they showed that the cumulative size distribution of SNRs is related to the ambient density distribution. The dependence of radio emission on density was explored in hydrodynamic simulations presented by [16]. Their results show a dependence on density, which may be too complex to capture in simple analytic models.

In summary, the current study considers a large sample of 54 SNRs in the Galaxy which have X-ray spectra and measured radio flux densities. We obtain their physical properties from the X-ray spectra using evolutionary models. For the first time, this enables a comparison of analytic radio emission models to data.

The SNR properties including shock velocity, radius and density are input into the recent model of S2017, to obtain the model radio emission. The resulting model predictions are compared to observations to find that the model is insufficient. Then we explore alternate power-law forms for the dependence of the radio emission on shock velocity, density and shock radius, to find an empirical analytic radio emission model. The empirical model is significantly better than the original radio emission model; however, it does not predict well the observed radio emission. This indicates that other factors are important for the radio emission, such as variations in the ISM magnetic field or in emission process, that are not dependent on the SNR properties derived from SNR X-ray emission.

**Author Contributions:** Conceptualization, D.A.L. and M.F.; methodology, D.A.L.; software, D.A.L. and F.M.; analysis, D.A.L. and F.M.; writing—original draft preparation, F.M. and D.A.L.; writing—review and editing, D.A.L. and M.F.; funding acquisition, D.A.L. All authors have read and agreed to the published version of the manuscript.

**Funding:** This research was funded by the Natural Sciences and Engineering Research Council of Canada.

**Data Availability Statement:** Not applicable.

**Conflicts of Interest:** The authors declare no conflict of interest. The funders had no role in the design of the study; in the collection, analyses, or interpretation of data; in the writing of the manuscript; or in the decision to publish the results.

## Abbreviations

The following abbreviations are used in this manuscript:

| | |
|---|---|
| SN | supernova |
| SNR | supernova remnant |
| ISM | interstellar medium |
| CC | core collapse |

## Notes

[1]     The dependence of $\eta$ on explosion energy $E_0$, was tested: the result was a scatter diagram with essentially no correlation between $\eta$ and $E_0$, with correlation coefficient $r^2 = 0.07$

[2]     This was implemented by minimizing the sum of absolute values of the power-law indices for the power-law fits to $H$ vs. $v_s$ and $H$ vs. $n_0$.

[3]     The dependence of $J$ on energy $E_0$ was checked, and showed essentially no correlation, with correlation coefficient of 0.02.

[4]     We noticed a numerical error in Equation (A4) of [15]: we use the corrected value from [19].

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
