# Peer review of "Radio Emission from Supernova Remnants: Model Comparison with Observations"

_universe, doi:10.3390/universe8120653_

Round 1

Reviewer 1 Report

Referee report for Radio Emission from Supernova Remnants: Model vs. Observations 

by Denis Leahy Felicity Merrick and Miroslav Filipovic

In this work the authors derive an improved relation the luminosity of supernova remnants (SNR) and its shock velocity, shock radius and  interstellar medium density. The latter, a new element over the previous reported relation. Those physical quantities are derived from the observation of the detected X-ray spectra of 54 SNRs.

The study is clearly presented. It is well shown how the improved version provides better correlations of quantities which is to some extend theoretically justified by hydrodynamic simulations. Therefore, I would like to recommend this article for publication.

Author Response

Authors response: 
We thank the reviewer for reviewing our paper. 
The paper has been additionally revised to take into account the comments of the other 2 reviewers.

Author Response

Our responses to the reviewers comments are separated by rows of *******

Reviewer:

Supernova remnants (SNRs) are diffuse, expanding structures resulting from a supernova explosion. Studies of SNRs are important for astronomers, as they play a key role in the evolution of galaxies, the acceleration of galactic cosmic rays, and so on. 
The authors investigated the relation between the observed radio luminosities of SNRs and the physical properties of those SNRs. 
By fitting, they found that the previous model can be improved significantly by including an interstellar medium density dependence in the efficiency of particle acceleration and magnetic field amplification. 
The paper is written well, interesting and deserves to be published.
However, in current state, I cannot recommend it for publication, a substantial modification is suggested.

Comments:
(1) As to the title, such a big title "Radio Emission from Supernova Remnants: Model vs. Observations" is indeed expected, but the content is disappointing due to little work as well as innovations
**************************
Authors response: 
We have edited the title to be more descriptive of the work: "Radio Emission from Supernova Remnants: Model Comparison with Observations"
**************************
(2) As to the abstract, some new parts such as model comparison, SNR sample number (if possible), simulation and error analysis should be added to make up for the deficiencies of the article's innovative points.
**************************
Authors response: 
We have rewritten, and added more information to, the abstract.
**************************
(3) In the abstract, the necessity, highlight and significance of this work are not reflected.
**************************
Authors response: 
We add a summary of the main significance of the work to the abstract. 
**************************
(4) In the introduction, to attract more readers' attention, please add some work on the pulsars and their SNRs. For example, assuming the age of a pulsar is equal to its associated SNR's age, some authors studied pulsars' spin-down evolution and magneto-thermal evolutions (Gao et al. MNRAS, 2016, 456, 55; Gao et al. ApJ, 2017, 849 , 19;
Wang et al. 2020, Universe, 6, 63 ;Yan et al. 2021, AN, 342,249).
**************************
Authors response: 
We have expanded the introduction and included the above references.
**************************
(5) In the section 2, when calculating and simulating, the authors only used s=0 and n=7, in order to show the calculation clearly, please adding the following formula
$\rho_{ISM}=\rho_s r^{−s}$ and $\rho_{ej}\propto r^{−n}$
**************************
Authors response:
We have done this.
**************************
(6) In Figure 1, it is suggested to label the different types of SNRs separately. It might be a little clearer if you divide energy and density into two graphs.
**************************
Authors response:
We have done this.
**************************
(7) In the section 2, line 71, the authors used the energy spectrum index of 1GHz to calculate the energy flux density of 1.4GHz, the formula should be $S_{\nu}\propto \nu^{−\alpha}$. With the increase of frequency, if the spectral index is unchanged, the energy flux S should decrease, but the energy flux data in Table 1 is larger than the original data. Why?
**************************
Authors response:
Thank you for noticing the sign error.
We corrected the error, and redid all calculations with the correct (smaller) flux at 1.4 GHz.
**************************
Line 72, the SNRs with unknown energy spectrum index were calculated by using the average energy spectrum index of other SNRS. What causes for you did so?
**************************
Authors response:
For SNRs with unknown spectrum index, a value has to be assumed to correct the 1 GHz flux density to 1.4 GHz. The most reasonable value to use is the average spectral value for SNRs. This is preferable to no correction.
**************************
(8) In the section 3.2, lines 109-111, it is suggested to give a specific formula or description for the highlighted part in the attachment.
**************************
Authors response:
We give the formula now.
**************************
(9) In the section 3.3, line 117, please give a corresponding fit plot.
**************************
Authors response:
The plot is given by Fig.4, which is now referenced. 
**************************
(10) It is suggested to give the revised formula of L_{1.4}, and then compare the predicted value with the observed data, which will be more convincing.
**************************
Authors response:
We have done this now at the end of section 3.3. 
**************************
(11) In the discussion, the authors claimed that "This dependence on density has been confirmed in other models", and suggest that the relevant discussion be included in the introduction. At the end of the paper, the results can be compared with those of hydrodynamics simulation.
**************************
Authors response:
We have added a short discussion of the parameter (including density) dependence in hydrodynamic models to the introduction (second last paragraph).
**************************
Minor problems: the errors of the data in Table 1 are larger, and there are some spelling errors in the language.
**************************
Authors response:
We have fixed the errors in distance for Table 1 (previously we had listed upper and lower limits as +/- by mistake). We corrected the footnotes spelling/grammar.
**************************
The paper has been additionally revised to take into account the comments of the other 2 reviewers.

Reviewer 3 Report

Referee report on the paper of D.Leahy et al. "Radio Emission from Supernova Remnants: Model vs. Observations"

The authors investigate the relation between radio luminosity and physical parameters of supernova remnants (SNRs). List of 54 SNRs with physical parameters derived from X-ray observations is used. Comparing the observed radio  luminosity with predictions of analytical model (Sarbadhicary et al. 2017) the authors concluded that density dependence of magnetic amplification or electron acceleration efficiency are necessary to explain the radio observations. 

I find that the results of the paper about density dependence are interesting and recommend publication after a minor revision. 

Comments:

1) The authors confuse quantities in Eq. 2. The second term contains $\mu G$ that is a microGauss but not the gravitational constant $G$ and proton mass $\mu $. This term is inverse Alfven Mach number  (see Eq. ( A7) of Sarbadhicary et al. (2017)) that is the ratio of Alfven speed to the shock speed. 

2) An important result is that density dependence of magnetic amplification or electron acceleration is necessary to explain the radio observations. This dependence absents in Eq. (2) taken from Sarbadhicary et al. (2017). They in turn refer to Crutcher (1999) to relate magnetic field strength and density. However this result was obtained for high densities >100 in molecular clouds where the gas motions  amplify the field in such a manner that Alfven speed is constant. Hardly this is a good assumption for lower density in interstellar medium.  Probably a better assumption is density independent magnetic field (say 3-5 microG) when its pressure is comparable with the interstellar pressure. In this case density dependence (inverse square root) will appear in Eq. 2. Probably the rest is related with density dependence of the electron acceleration efficiency. It is difficult to justify this theoretically, but we know from observations of SNRs that cosmic ray electron to proton ratio is lower in young SNRs. Probably electron acceleration efficiency is somehow proportional to the ratio of Alfven or sonic speed to the shock speed. For density independent gas or magnetic pressure this would give an additional density and shock speed dependence. 

Author Response

Our responses to the reviewers comments are separated by rows of *******

Reviewer:

The authors investigate the relation between radio luminosity and physical parameters of supernova remnants (SNRs). List of 54 SNRs with physical parameters derived from X-ray observations is used. Comparing the observed radio  luminosity with predictions of analytical model (Sarbadhicary et al. 2017) the authors concluded that density dependence of magnetic amplification or electron acceleration efficiency are necessary to explain the radio observations. 

I find that the results of the paper about density dependence are interesting and recommend publication after a minor revision. 

Comments:

1) The authors confuse quantities in Eq. 2. The second term contains $\mu G$ that is a microGauss but not the gravitational constant $G$ and proton mass $\mu $. This term is inverse Alfven Mach number  (see Eq. ( A7) of Sarbadhicary et al. (2017)) that is the ratio of Alfven speed to the shock speed. 
**************************
Authors response:

Thank you for pointing out the error in \mu G. We have redone our calculations. 
Some details are supplied here: 

We used equation (A7) of Sarbadhicary+2016, with 
$M_A= v_s/v_a= v_s sqrt(4\pi\rho_0)/B_0$ and 
$B_0 = 9\mu G(\rho_0/1.6\times 10^-{27}g cm^{-3})^{0.47}$. 
Our previous error using G instead of microgauss (6.67e-8*mu vs. 1e-6 is a factor of 6.67e-2*mu.
However S2017 also made an error (which we found by comparison with the Crutcher 1999 paper): they used 1.6e-27 in their equation (A4): the correct value deduced from the Crutcher paper is 1.6e-24 (the error is perhaps due to a units conversion of SI to cgs). This factor of 1000 change is in the square root (sqrt(1000)=31.6), so the net error we made was 31.6*6.67e-2*mu=2.1*mu.

We have put the correct equations and calculations for \epsilon_b in the paper
and revised the calculations throughout. (It does not affect Fig.3, which did not use \epsilon_b). 
**************************

2) An important result is that density dependence of magnetic amplification or electron acceleration is necessary to explain the radio observations. This dependence absents in Eq. (2) taken from Sarbadhicary et al. (2017). They in turn refer to Crutcher (1999) to relate magnetic field strength and density. However this result was obtained for high densities >100 in molecular clouds where the gas motions  amplify the field in such a manner that Alfven speed is constant. Hardly this is a good assumption for lower density in interstellar medium.  Probably a better assumption is density independent magnetic field (say 3-5 microG) when its pressure is comparable with the interstellar pressure. In this case density dependence (inverse square root) will appear in Eq. 2. Probably the rest is related with density dependence of the electron acceleration efficiency. It is difficult to justify this theoretically, but we know from observations of SNRs that cosmic ray electron to proton ratio is lower in young SNRs. Probably electron acceleration efficiency is somehow proportional to the ratio of Alfven or sonic speed to the shock speed. For density independent gas or magnetic pressure this would give an additional density and shock speed dependence. 
**************************
Authors response:

We agree that the Crutcher result is derived from molecular cloud observations for densities >100 cm^{-3}. 
Their relation is B\propto \sqrt(\rho) (within errors as their fit index is 0.47\pm0.08). The same relation is obtained whenever equipartition applies between magnetic field pressure and gas pressure.
The simple relation B\propto \sqrt(\rho) may not hold at low densities and magnetic field values in the Galaxy needs more investigation.

We add some text in the discussion section of the paper regarding this point the reviewer has made. 
**************************
The paper has been additionally revised to take into account the comments of the other 2 reviewers.

Round 2

Reviewer 2 Report

According to the comments and suggestions in my first report, the authors have made substantial modifications for this revised manuscript. I would like to recommend it for publication. Best wishes Referee